# Neutralization of Different Variants of SARS-CoV-2 by a F(ab′)2 Preparation from Sera of Horses Immunized with the Viral Receptor Binding Domain

**DOI:** 10.3390/antib12040080

**Published:** 2023-12-07

**Authors:** Mariajosé Rodriguez-Nuñez, Mariana del Valle Cepeda, Carlos Bello, Miguel Angel Lopez, Yoneira Sulbaran, Carmen Luisa Loureiro, Ferdinando Liprandi, Rossana Celeste Jaspe, Flor Helene Pujol, Héctor Rafael Rangel

**Affiliations:** 1Laboratorio de Virología Molecular, Centro de Microbiología y Biología Celular, Instituto Venezolano de Investigaciones Científicas, Caracas 1020, Venezuela; rodriguez95mariajose@gmail.com (M.R.-N.); yfsulbara@gmail.com (Y.S.); cloureir@gmail.com (C.L.L.); rossanajaspesec@gmail.com (R.C.J.); 2Biotecfar S.A., Facultad de Farmacia, Universidad Central de Venezuela, Caracas 1050, Venezuela; marianacepedab@gmail.com (M.d.V.C.); carlosbiotecfar@gmail.com (C.B.); queregua@hotmail.com (M.A.L.); 3Laboratorio de Biología de Virus, Centro de Microbiología y Biología Celular, Instituto Venezolano de Investigaciones Científicas, Caracas 1020, Venezuela; fliprand@gmail.com

**Keywords:** COVID-19, variants, equine sera, neutralizing antibodies, immune escape, evolution

## Abstract

The Receptor Binding Domain (RBD) of SARS-CoV-2, the virus responsible for the COVID-19 pandemic, is the functional region of the viral Spike protein (S), which is involved in cell attachment to target cells. The virus has accumulated progressively mutations in its genome, particularly in the RBD region, many of them associated with immune evasion of the host neutralizing antibodies. Some of the viral lineages derived from this evolution have been classified as Variant of Interest (VOI) or Concern (VOC). The neutralizing capacity of a F(ab′)2 preparation from sera of horses immunized with viral RBD was evaluated by lytic plaque reduction assay against different SARS-CoV-2 variants. A F(ab′)2 preparation of a hyperimmune serum after nine immunizations with RBD exhibited a high titer of neutralizing antibodies against the ancestral-like strain (1/18,528). A reduction in the titer of the F(ab’)2 preparation was observed against the different variants tested compared to the neutralizing activity against the ancestral-like strain. The highest reduction in the neutralization titer was observed for the Omicron VOC (4.7-fold), followed by the Mu VOI (2.6), Delta VOC (1.8-fold), and Gamma VOC (1.5). Even if a progressive reduction in the neutralizing antibodies titer against the different variants evaluated was observed, the serum still exhibited a neutralizing titer against the Mu VOI and the Omicron VOC (1/7113 and 1/3918, respectively), the evaluated strains most resistant to neutralization. Therefore, the preparation retained neutralizing activity against all the strains tested.

## 1. Introduction

SARS-CoV-2 is the coronavirus responsible for the COVID-19 pandemic, which has caused at least 690 million cases and 6.9 million deaths worldwide [1]. In contrast with other RNA viruses, this viral family harbors a proofreading capacity, which limits the error induced by the RNA polymerase during replication. However, due to the enormous replication cycles that this virus has experienced during the pandemic, in addition to a high frequency of recombination and the effect of host editing enzymes, the mutation rate in the SARS-CoV-2 genome has been estimated at around 9.9 × 10^−4^ to 2.2 × 10^−3^ [2,3]. The high frequency of mutations has allowed the selection of variants with higher fitness, transmission capacity, and particularly evasion of the immunity present in the human host during the successive waves of infection [4].

The Receptor Binding Domain (RBD) of SARS-CoV-2 is the functional region of the viral Spike protein (S) involved in cell attachment to target cells through the ACE2 receptor. S can be divided into two regions: S1, which contains the RBD, and S2, which harbors a furin-cleavage site and a hydrophobic domain. Cleavage of the protease-sensitive site leads to exposure of the hydrophobic domain, allowing the fusion of the viral and the endosomal cellular membrane [5]. S has been the main target of vaccines not involving the whole inactivated virus, but RBD has also been tested for the design of some prototype vaccines [6]. Neutralizing antibodies operate mainly by preventing the interaction of the RBD with the ACE2 receptor. Mutations on the Spike protein may be associated with changes in the RBD affinity to cellular receptors [7], as well as with cellular tropism. The accumulation of mutations in S, and particularly RBD, can also affect the binding of the neutralizing antibodies produced during immunization (by natural infection or vaccination), allowing new variants to be less sensitive to neutralization but not totally resistant [4]. The emergence of variants had a great impact on the efficacy of the vaccines, allowing the variant viruses to infect the host despite the pre-existing immunity, vaccination retaining, however, the ability to prevent the severe clinical presentation of the disease [8].

Five VOCs of SARS-CoV-2 were recognized since the end of 2020 [9]. The first VOC described was the Alpha (original lineage B.1.1.7), which emerged in the U.K. [10]. The second VOC identified was the Beta (original lineage B.1.351), which emerged in South Africa [11]. The Gamma VOC (original lineage B.1.1.28.2, P.1) was the third designated VOC, originating in Brazil [12]. In April 2021, the Delta VOC (original lineage B.1.617.2) emerged in India [13,14]. The last and only VOC circulating at present is Omicron (original lineage B.1.1.529), which emerged in South Africa and harbored a huge number of mutations [15]. In addition to VOCs, the WHO also classified some lineages as Variants of Interest (VOIs), variants similar to VOC but for which the enhanced transmissibility or immune evasion was not necessarily confirmed [9]. Among these VOIs, the two most important emerged in South America: Lambda and Mu. The Lambda VOI (original lineage C.37) emerged probably in Peru [16] and the Mu VOI in Colombia (original lineage B.1.621) [17].

Hyperimmune equine plasma-derived fraction is widely used for snakebite envenoming. The worldwide annual incidence of snake bites is around 5 million cases, causing up to 150,000 deaths; equine-derived polyclonal antibodies are one of the best therapeutic agents against this global threat [18]. This immune therapy has also shown suitable efficacy against several viral infectious diseases, such as rabies [19], Ebola [20], and Middle East Respiratory Coronavirus Syndrome [21]. However, it is associated with several adverse effects due to the immune recognition of foreign antibodies. It has been shown that the F(ab′)2 part of polyclonal antibody of the hyperimmune serum exhibits a superior therapeutic effect, with lower side effects compared to the total serum [22].

The production of equine sera against venoms of snakes and scorpions, but also against several infectious agents, laid the foundations for producing an anti-SARS-CoV-2 equine serum [23,24,25,26,27,28]. Several viral antigens have been evaluated for the production of this biological: inactivated SARS-CoV-2 whole virus [23,24], S [25], or RBD [26,27,28]. A living systematic review of randomized controlled trials suggests that RBD-specific polyclonal F(ab′)2 fragments of equine antibodies may reduce mortality and serious adverse events and may reduce clinical worsening, at least before widespread vaccination against this virus [29].

In a previous study, we described that repeated immunizations were undertaken to produce hyperimmune F(ab′)2 preparations with high titers of antibodies against RBD [27]. The aim of this study was to evaluate the neutralizing activity of this hyperimmune serum against different variants of SARS-CoV-2.

## 2. Materials and Methods

### 2.1. Equine Anti-RBD Hyperimmune Preparation

The production and purification of a F(ab′)2 preparation anti-RBD equine serum was previously described [27]. This research has been approved by the Bioethics Committee of Hospital Universitario de Caracas in an Ordinary Meeting via online N° 06 dated 22 December 2020, following the norms obtained from the ARRIVE guidelines and was carried out in accordance with the U.K. Animals (Scientific Procedures) Act, 1986, and associated guidelines, EU Directive 2010/63/EU for animal experiments. Trained staff organized all the experimental methods relating to the use of live animals. Three horses were immunized with RBD antigen (Acro Biosystem, South Croydon, U.K.), as stated in Table 1. Three different F(ab′)2 preparations, a product of the mixture of the 3 horse sera corresponding to different time points, were evaluated at different dilutions. The first preparation was obtained at week 3, after 3 immunizations with low doses of antigen. The second preparation was collected at week 7, after 5 immunizations, and the third one at week 27, after 9 immunizations. The last immunizations were administered with a higher dose of antigen (Table 1).

The sera collected by bleeding of the animals was treated at 56 °C for one hour. The sera were centrifuged at 2000 rpm for 15 min and then passed through a 0.45 µm filter. The pooled sera were then precipitated with ammonium sulfate. For the preparation of F(ab′)2, The preparation was treated with pepsin (1.25 g/L) at 30 °C for 30 min to eliminate the Fc fragment from total IgG. After the precipitation of contaminating proteins, the preparation was filtered, and the pH of the filtrate was adjusted for a second precipitation with ammonium sulfate to recover the F(ab′)2 proteins. This preparation was diafiltered with distilled water before evaluation, and the protein was stored at 4 °C until use. The antibody titer of the preparations was determined by ELISA, using the same RBD antigen used for immunization. The immune reactivity was also tested by western immunoblotting. The absence of toxicity of the F(ab′)2 preparation was tested by inoculation of NIH mice; the animals did not show signs of clinical toxicity after 7 days of injection [27].

### 2.2. Viral Strains and Cells

Vero-E6 cells (ATCC No. CRL-1586) were kindly donated by the “Instituto Nacional de Higiene Rafael Rangel”, Caracas, Venezuela, adapted and maintained in RPMI 1640 medium supplemented with 10% FBS and antibiotic/antimycotic. Cells were passed every 3–4 days based on the monolayer confluence. The viruses were isolated in a BSL3 facility from nasopharyngeal swabs of patients who were positive (Ct below 25) between July 2020 and January 2022. For infection, the Vero-E6 cells were incubated with 0.22 μm filtered transport medium of positive samples for one hour. All the cultures were monitored daily, and the new virus production was confirmed by cytopathic effect; then, the supernatant was tested by RTq-PCR (Sansure Biotech Inc., Changsha, China). For variant assignment, the complete genome sequence of each strain was performed by NGS, as previously described [30]. The different strains were titrated by serial dilution in order to use the same virus titer in each assay.

### 2.3. Neutralization Assay

Plates of 24 wells were seeded with 200,000 cells/well and incubated overnight. The cells were infected with a mixture of each virus previously incubated at 37 °C for 30 min with different dilutions of anti-RBD F(ab′)2 preparation. Then, a volume of this mix, calculated to produce approximately 80 plaques, was inoculated to the cells in triplicate wells. The infection process was performed for one hour at 37 °C in a 5% CO_2_ atmosphere, and the cells were washed twice with PBS to retire all the non-internalized viruses. Then, the cells were overlayed with 0.5% carboxymethyl cellulose in a culture medium [31]. The plates were incubated for 72 h, fixed with 4% formaldehyde, and stained with crystal violet to count the number of lytic plaques under each condition. Neutralization titers were determined by logistic regression as the final dilution, producing a 50% reduction in the average number of plaques of the controls (GraphPad Software Inc., La Jolla, CA, USA).

## 3. Results

In order to test the neutralization ability of the F(ab′)2 preparation obtained from equine hyperimmune sera, several lineages of SARS-CoV-2 were successfully cultivated and used for neutralization assays (Table 2). The different isolated viruses were B.1.1.33 (equivalent to the ancestral virus), P.1 (Gamma VOC), AY122 (Delta VOC), B.1.621 (Mu VOC), and BA.1.1 (Omicron VOC). It is interesting to note that for all the lineages except Omicron, only one isolate was necessary to succeed in obtaining the viral culture used for the assays. For the Omicron lineage, three isolates were tested to finally obtain the viral culture. The isolated successfully cultivated was from a non-vaccinated patient.

Figure 1 shows the amino acid sequence of each of the lineages used in this study, together with the sequence used for producing the RBD immunogen. Three variants exhibit mutations in amino acid 484, associated with an important reduction in antibody binding to the RBD: P1, Mu (E484K), and Omicron (E484A). In addition, this last variant harbors several other mutations involved, at different intensities, in the reduction of antibody binding; of note, the Y505H mutations have also been associated with an important reduction in antibody binding (Figure 2). The amino acid sequences of the P1 and Mu variants are very similar; the main differences are the presence of the mutation K417T in the P1 variant and of R346K in the Mu one. Both mutations have been associated with a low-intensity reduction in polyclonal antibody binding (Figure 1).

F(ab′)2 preparations from sera sampled at different time points (Table 1) were evaluated. Table 3 shows the titer of each preparation able to reduce in 50% the number of lytic plaques against each of the lineages tested (IC50). No neutralization was observed at 1/50 dilution for an unrelated antiophidic F(ab′)2 preparation. No neutralization was observed at 1/50 dilution for F(ab′)2 preparation 001; this dilution was able to reduce only between 30 and 45% the number of lytic plaques of the different strains, not reaching 50% reduction for any of them. The 1/200 dilution of the F(ab′)2 preparation 002 led to a reduction in lytic plaques between 52 and 76% for the different variants. The enrichment in neutralizing antibodies was notorious for the F(ab′)2 preparation 003 (Table 3). This F(ab′)2 preparation exhibited a high titer of neutralizing antibodies against the ancestral-like strain (1/18,528). A reduction in the titer of the F(ab′)2 preparation was observed against the different variants tested. The highest reduction was observed for the Omicron VOC (4.7-fold), followed by the Mu VOI (2.6), Delta VOC (1.8-fold), and Gamma VOC (1.5). Even if a progressive reduction in the neutralizing antibodies titer against the different variants evaluated was observed, the serum still exhibited a significant neutralizing titer against the Mu VOI and the Omicron VOC, the strains evaluated to be most resistant to neutralization.

Figure 2 shows the percentage of neutralization for each dilution of the F(ab′)2 preparation 003 against each variant. A reduction in the neutralizing capacity of the F(ab′)2 preparation 003 was observed at every dilution for each variant, compared to the neutralization exhibited against the ancestral-like strain, although not at the same intensity. The highest reduction in the percentage of neutralization was observed for the Omicron variant, followed by the Mu, Delta, and P1 ones (Figure 2).

## 4. Discussion

The knowledge accumulated since 2003 with the emergence of SARS-CoV stressed the importance of the RBD in triggering the production of strong neutralizing antibodies [33]. A study conducted on 647 patients infected with SARS-CoV-2 indicated that around 90% of the neutralizing antibodies target the RBD region. The immunodominance of the RBD could be associated with its low level of glycosylation and its higher accessibility compared to the rest of S [34]. Nevertheless, it is known that other regions of S contribute to inducing a protective immune response against SARS-CoV-2, such as the N-terminal region of S1 [35,36].

The equine F(ab′)2 preparations analyzed in this study were obtained with immunization with the RBD of SARS-CoV-2. Although the animals might have been exposed previously to equine coronavirus [27], the only source of the neutralizing antibodies found in the F(ab′)2 preparation should be the commercial RBD used as an immunogen. This allows for evaluating the immunogenicity of this region in inducing neutralizing antibodies against SARS-CoV-2 out of the context of the whole S protein and eliminating the participation of other regions of this protein. In our previous study, a high titer of antibodies against RBD was detected by ELISA (more than 1/24,000) after 3 immunizations [27]. However, as shown here, this high titer was not accompanied by a high titer of neutralizing antibodies in the first F(ab′)2 preparations (Table 3). A great number of immunizations and a higher dose of the immunogen were needed to induce a significant level of neutralizing antibodies, as shown by the absence of a minimal neutralizing activity of F(ab′)2 preparations 001 and 002, respectively. In contrast, after 15 immunizations with higher doses of RBD (Table 1), the neutralization titer of the F(ab′)2 preparation 003 was quite important.

Since the end of 2020 until now, the waves of the SARS-CoV-2 epidemic have been characterized by the emergence of VOCs, generally disseminated all around the world. Each VOC showed different mutation patterns in different regions of the genome; many of them occurred in the RBD, and precisely these mutations are the ones that can confer increased infectivity to the viruses, increase the receptor binding process, and evasion of neutralizing antibodies [37]. In this study, we tested the neutralizing ability of the F(ab′)2 preparations against 4 VOCs and one VOI. As expected, a significant reduction in the titer of neutralizing antibodies was found for the F(ab′)2 preparation 003 (the one exhibiting a significant neutralizing activity) against all the variants when compared to the activity against the ancestral strain. The highest reduction in neutralizing activity was observed against the Omicron variant, in agreement with previously reported characteristics of this VOC to exhibit a great number of mutations associated with immune escape [15,32].

The second variant exhibiting the highest reduction in neutralizing activity of the equine preparation was indeed the VOI Mu. It is interesting to note that the RBD sequence of the Mu VOI is very similar to the one of the P1 VOC. Both variants harbor the mutation E484K, which has been associated with a strong reduction in the binding of neutralizing antibodies: the Mu VOI lacks the mutation K417T characteristic of the P1 VOC but instead harbors the R346K mutation. K417T and R346K mutations have been graded similarly in their contribution to immune evasion (Figure 1) [4,32]. A study of free energy perturbation predicts that the reduction of antibody binding caused by the R346K mutation (Mu) might be even lower than the one caused by the K417N one (present in the Beta VOC, not tested in our study) [38]. However, the R346K mutation is also present in the strain used in our study, sub-lineage BA.1.1 of the Omicron VOC; this mutation has been associated with immune evasion.

Therapeutic equine polyclonal antibodies from Costa Rica were assessed against several variants, including P1 and Delta VOCs. The authors found similar IC50 concentrations in their preparations and a similar increase in IC50 when testing P1 and delta VOCs. In their case, the animals were immunized with the S1 region of the S protein, which includes RBD, or with all the structural viral proteins. The authors did not test the Mu VOI nor the Omicron VOC; the latter had not yet emerged when this study was conducted [39]. Equine F(ab′)2 preparation produced by immunization with the whole inactivated virus proved to bind to wildtype and all the VOCs RBD by surface plasmon resonance experiments [23]. However, similar reductions in the neutralization ability against the different viral VOCs were observed, irrespective of the antigen used for immunization, i.e., when using the whole inactivated virus [24] or the S protein [26]. None of these studies evaluated the neutralization of their biological preparation against the Mu VOI.

In a study from Japan, the neutralization titer of sera from vaccinated and convalescent patients was tested against seven variants: Alpha, Beta, Gamma, Delta, Epsilon, Lambda, and Mu, but not Omicron, which emerged after the time of their study. The Mu VOI was the one that elicited the highest reduction in neutralization titer [40], in agreement with the observation of our study. A similar reduction in neutralization ability against the Mu VOI was observed among vaccinated individuals in Colombia [41]. These results suggest that this variant, which did not reach the classification of VOC, indeed exhibits a high ability to evade the immune response induced by the ancestral strain [42].

## 5. Conclusions

The immunization with several boosters of high doses of SARS-CoV-2 RBD led to the production of a F(ab′)2 preparation with high titers of neutralizing antibodies against the ancestral viral strain. As expected, this titer was reduced against some of the variants, particularly the Omicron VOC and, interestingly, against the Mu VOI. However, the preparation retained neutralizing activity against all the strains tested.

## Figures and Tables

**Figure 1 antibodies-12-00080-f001:**
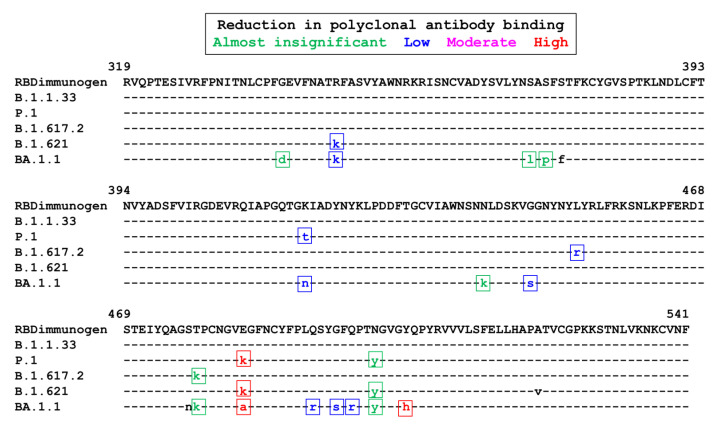
Amino acid alignment of the RBD used for immunization and of the different lineages tested. The boxes and colors indicate the mutations involved in the reduction of binding of polyclonal antibodies, as previously reported [4,32].

**Figure 2 antibodies-12-00080-f002:**
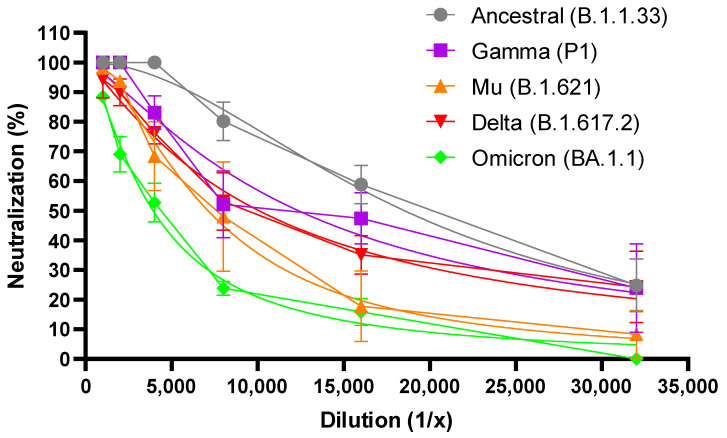
Percent neutralization of the F(ab′)2 preparation 003 of the different variants of SARS-CoV-2. Results from two independent experiments are shown, each point being the mean value (± and standard deviation) of 3 replicas.

**Table 1 antibodies-12-00080-t001:** Timeline of immunization and bleeding of horses.

Week	RBD Dose (µg)	Adjuvant	F(ab′)2 Preparation	F(ab′)2 Concentration (mg/mL)
0	100	Comp. Freund ^1^		
1	200	Inc. Freund ^1^		
2	300	PBS ^2^		
3			001	14.8
4	600	PBS		
6	600	PBS		
7			002 ^3^	11.2
10	1000 ^4^	PBS		
12	1000 ^4^	PBS		
20	1000 ^4^	PBS		
26	1000 ^4^	PBS		
27			003 ^5^	11.9

^1^ Complete and incomplete Freund adjuvant. ^2^ Phosphate Buffer Saline (PBS). ^3^ Preparation 002 is the product of the bleeding at 3 weeks plus a remnant of preparation 001. ^4^ The horses were immunized with 500 µg of RBD in a two-day interval. ^5^ Preparation 003 is a mixture of several bleedings after preparation 002.

**Table 2 antibodies-12-00080-t002:** Viral strains used for neutralization assays.

Lineage	Variant	Name	GISAID Accession Number	Collection Date
B.1.1.33	Ancestral	CM1-6AV	EPI_ISL_6980947	12 July 2020
P1	Gamma	Dtt54	EPI_ISL_2628299	13 March 2021
B.1.621	Delta	TacMa	EPI_ISL_6976265	29 July 2021
AY122	Mu	MirLab4	EPI_ISL_9486877	16 August 2021
BA.1.1	Omicron	CULT-OM	EPI_ISL_17389567	11 February 2022

**Table 3 antibodies-12-00080-t003:** IC50 neutralization of each F(ab′)2 preparation against different variants.

F(ab′)2 Preparation ^1^	Ancestral	P1	Variant Delta	Mu	Omicron
001	<1/50 ^2^	<1/50	<1/50	<1/50	<1/50
002	1/533	1/339	1/318	1/181	1/123
003	1/18,528	1/12,364	1/10,071	1/7,113	1/3,918
003 (µg/mL) ^3^	0.64	0.96	1.18	1.67	3.04

^1^ Each F(ab′)2 preparation was from equine sera after 3 (001), 5 (002), and 9 (003) immunizations with the RBD of SARS-CoV-2, as described in Table 1. ^2^ Titers correspond to the dilution of the preparation causing a reduction of 50% in the number of lytic plaques. Each titer was obtained by logistic regression of the average of 3 replicas and is the geometric mean value of two independent experiments. ^3^ Protein concentration of the preparation 003 able to reduce 50% of the lytic plaques.

## Data Availability

The complete genome sequences have been deposited in the GISAID database (accession numbers EPI_ISL_6980947, EPI_ISL_2628299, EPI_ISL_6976265, EPI_ISL_9486877, EPI_ISL_17389567).

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
