# Peer review of "Neutralization of Different Variants of SARS-CoV-2 by a F(ab′)2 Preparation from Sera of Horses Immunized with the Viral Receptor Binding Domain"

_2073-4468, 2023, doi:10.3390/antib12040080_

Round 1
Reviewer 1 Report
Comments and Suggestions for Authors
This was a nicely presented paper. It would be of interest to include a little more context about equine immunoglobulin for passive immunotherapy.
I had several questions about Table 1. Concentration in mg/mL of what? F(ab')2 prep? Please explain 001 -- is this a pre-bleed? more details on 002 and 003 please ? Are these collected after different numbers of immunizations? timepoints?
Please make clear that the reduction in neutralization titer was in comparison to the ancestral strain titer.
I'm confused about the ELISA results included in the discussion. Is this your research? If so, please include in methods and results as well.
there is a reference #34 that is not included in the ref list.
Comments on the Quality of English LanguageThe English could use some editing. You can take out the word "ones" after the variant names. "titer" not "title" in your abstract. Take out "during immunization" and remove parentheses on line 52. Check ?, line 97. isolated= isolate line 127. I will leave the rest up to the editors of this journal!
Author Response
Reviewer 1
This was a nicely presented paper. It would be of interest to include a little more context about equine immunoglobulin for passive immunotherapy.
Thank you very much for your comment. More context about equine immunoglobulin for passive immunotherapy was included in the Introduction, lines 70-78 and 80-83.
I had several questions about Table 1.
We are sorry for the misleading information.
Concentration in mg/mL of what? F(ab')2 prep?
Yes. This information was included in Table 1.
Please explain 001 -- is this a pre-bleed?
more details on 002 and 003 please?
Are these collected after different numbers of immunizations? Time-points?
Preparation 001 is not a pre-bleed. The Table was corrected. More details were included in lines 99-101.
Please make clear that the reduction in neutralization titer was in comparison to the ancestral strain titer.
We apologize for the omission. This is now included in lines 23 (Abstract) and192-193.
I'm confused about the ELISA results included in the discussion. Is this your research? If so, please include in methods and results as well.
We apologize for the confusion. The ELISA results were from a previous study. This information was clarified in lines 217, 219 and 220.
there is a reference #34 that is not included in the ref list.
We are sorry for the mistake in citing the references, which is now corrected. This reference is now reference 41.
Comments on the Quality of English Language
The English could use some editing. You can take out the word "ones" after the variant names. "titer" not "title" in your abstract. Take out "during immunization" and remove parentheses on line 52. Check ?, line 97. isolated= isolate line 127. I will leave the rest up to the editors of this journal!
Thank you very much for your comments. We edited the manuscript accordingly and also checked the whole manuscript for English accuracy.
Reviewer 2 Report
Comments and Suggestions for Authors
This is very interesting. While it was expected that the changes in the virus would allow it to evade acquired immunity, it is very important that the data are actually presented.
I would have liked to see the results of more recent Omicron variants if possible, but I guess we should be thankful for the coverage up to BA.1.1, otherwise publication would have been delayed.
Author Response
Reviewer 2
This is very interesting. While it was expected that the changes in the virus would allow it to evade acquired immunity, it is very important that the data are actually presented.
Thank you very much for your comment.
I would have liked to see the results of more recent Omicron variants if possible, but I guess we should be thankful for the coverage up to BA.1.1, otherwise publication would have been delayed.
Thank you very much for your comment. Unfortunately, the BA.5 isolate was not available for this study.